# Diagnostic Role of Four-Dimensional Computed Tomography for Preoperative Parathyroid Localization in Patients with Primary Hyperparathyroidism: A Systematic Review and Meta-Analysis

**DOI:** 10.3390/diagnostics11040664

**Published:** 2021-04-07

**Authors:** Lixin Sun, Jian Yao, Pan Hao, Yuanyuan Yang, Zhimou Liu, Ruchen Peng

**Affiliations:** 1Department of Nuclear Medicine, Beijing Luhe Hospital, Capital Medical University, No 82 Xinhua South Road, Tongzhou District, Beijing 101149, China; slxbjmu@163.com (L.S.); swordyao@126.com (J.Y.); xiaopan217511@163.com (P.H.); y352827441@163.com (Y.Y.); liuzhimou@mail.ccmu.edu.cn (Z.L.); 2Department of Radiology, Beijing Luhe Hospital, Capital Medical University, No 82 Xinhua South Road, Tongzhou District, Beijing 101149, China

**Keywords:** primary hyperparathyroidism, 4D-CT, localization, diagnostic accuracy, meta-analysis

## Abstract

We sought to systematically evaluate diagnostic performance of four-dimensional computed tomography (4D-CT) in the localization of hyperfunctioning parathyroid glands (HPGs) in patients with primary hyperparathyroidism (pHPT). We calculated the pooled sensitivity, specificity, positive likelihood ratio (PLR), negative likelihood ratio (NLR), and diagnostic odds ratios (DOR) of 4D-CT on a per-lesion level, as well as pooled sensitivity and positive predictive value (PPV) on a per-patient level with 95% confidence intervals (CIs). Additionally, we plotted summary receiver operating characteristic (SROC) curves and evaluated the areas under the curves (AUC). A total of 16 studies were included in the analysis. Their pooled sensitivity, specificity, PLR, NLR, and DOR of 4D-CT on per-lesion level were 75% (95%CI: 66–82%), 85% (95%CI: 50–97%), 4.9 (95%CI: 1.1–21.3), 0.30 (95%CI: 0.19–0.45), and 17 (95%CI: 3–100), respectively, with an AUC of 81% (95%CI: 77–84%). We also observed heterogeneity in sensitivity (I^2^ = 79%) and specificity (I^2^ = 94.7%), and obtained a pooled sensitivity of 81% (95%CI: 70–90%) with heterogeneity of 81.9% (*p* < 0.001) and PPV of 91% (95%CI: 82–98%) with heterogeneity of 80.8% (*p* < 0.001), based on a per-patient level. Overall, 4D-CT showed moderate sensitivity and specificity for preoperative localization of HPG(s) in patients with pHPT. The diagnostic performance may improve with 4D-CT’s promotion to first-line use on a lesion-based level, further research is needed to confirm the results.

## 1. Introduction

Primary hyperparathyroidism (pHPT) is a common endocrine disorder, defined as hypercalcemia secondary to excessive secretion of parathyroid hormone by 1 or more hyperfunctioning parathyroid glands (HPGs) [1,2]. It more commonly affects elderly individuals, and women four times as often as men [3,4]. A single parathyroid adenoma (85–90%) is the most common cause of pHPT [5]. Parathyroidectomy (PTx) represents the best line of action for management of parathyroid adenoma(s). It is advised for patients with systematic pHPT and those who meet the surgery guideline criteria. On the other hand, it is always an option for some patients with asystematic pHPT, even if they do not meet any criteria for surgery [6,7]. In addition, focused parathyroidectomy and bilateral neck exploration (BNE) have been shown to result in analogous biochemical complications and cure rates. Nevertheless, focused parathyroidectomy has some advantages, including decreased operating time, length of hospital stay as well as reduced medical cost and increased patient comfort [8]. However, clinicians need to be preoperatively precise in the localization of HPG(s) before surgery [5].

Although neck ultrasound (US) and 99mTc-sestamibi single-photon emission computed tomography/computed tomography (SPECT/CT) are the most commonly used modalities for localizing HPG(s), parathyroid localization has recently undergone considerable technological advancements [7]. Four-dimensional computed tomography (4D-CT), first reported in 2006, has emerged as a promising imaging modality for preoperative localization in patients with pHPT [9]. Summarily, 4D-CT is a contrast enhanced multiple-phase CT that comprises three or four common phases, namely non-contrast-enhanced, arterial and delayed phase. The first three dimensions entail axial acquisition with coronal and sagittal reformations, whereas the fourth “dimension” involves change in enhancement over time from the multiple-phase image. Parathyroid adenoma(s) always show low attenuation on the non–contrast enhanced imaging, rapid and peak enhancement on the arterial phase, then washout of contrast from arterial to delayed phase [10]. In the past decade, several studies have demonstrated 4D-CT’s diagnostic accuracy in localizing HPG(s) in patients with hyperparathyroidism (HPT), thereby to a correct side and/or quadrant on per-patient and/or per-lesion level, albeit with a wide range of sensitivities (ranging from 50–100%) and specificities (ranging from 0–100%) with number of patients ranging from 33–400 [11,12,13]. Additionally, previous meta-analyses found that 4D-CT exhibited a pooled sensitivity of 89.4%, 0.85 (95%CI, 0.69–0.94) and specificity of 0.93 (95%CI, 0.88–0.96), respectively [14,15]. However, the authors only enrolled two and nine studies, necessitating further investigation.

The present study aimed to evaluate diagnostic performance of 4D-CT for the localization of HPG(s) using a meta-analysis. The findings are expected to guide its application quadrant in patients with pHPT on both lesion-based and patient-based basis.

## 2. Materials and Methods

This systematic review and meta-analysis was performed according to the Preferred Reporting Items for Systematic Reviews and Meta-Analyses (PRISMA) guidelines, nevertheless, it was not registered on the international prospective register of systematic reviews (PROSPERO) [16].

### 2.1. Search Strategy

We searched electronic literature databases, namely PubMed, Embase, and Web of science, from their inception up to 10th November 2020. The search algorithm (Appendix A) was based on the following combined terms: (a) “Four-dimensional computed tomography or 4DCT or 4D-CT or Four dimensional computed tomography”, (b) “hyperparathyroidism or (parathyroid adenoma)”, and (c) “(diagnostic accuracy) or sensitivity or specificity”. No start date restrictions were included in our search.

### 2.2. Inclusion Criteria and Exclusion Criteria

Studies that met the following criteria were included in the analysis: (1) patients with primary hyperparathyroidism undergoing 4D-CT for localization of HPG(s); (2) surgery and histology as the gold standard; (3) the outcome was diagnostic accuracy expressed as sensitivity and specificity on lesion-based basis and sensitivity on patient-bases basis; and (4) reported sensitivity and/or specificity for a correct quadrant or typical parathyroid sites or in ectopic areas.

Articles were excluded if they were duplicate publications, case reports, abstracts, review articles, conference/meeting papers, lacked full text or were not written in English. Moreover, articles that reported insufficient data to reassess sensitivity and specificity as well as those that only described diagnosis of HPG(s) to a correct side were also excluded.

### 2.3. Data Extraction and Quality Assessment

Basic information, namely first author name, publication date, country, study design, and patients’ characteristics, as well as technical aspects, such as machine model, detector, product, injection rate, dose, scan phase, and imaging procedure, were grouped together. Each included study was analyzed to obtain the numbers of true positives (TP), false positives (FP), true negatives (TN), and false negatives (FN) of per-lesion or per-patient for localization of HPG(s). We applied the Quality Assessment of Diagnostic Accuracy Studies 2 (QUADAS-2) tool to assess studies quality. Each article was reviewed by two reviewers (L.S. and P.H.) and any discrepancies were resolved by discussion. The result was judged as true positive if 4D-CT localized the correct position of the HPG(s) (upper/lower pole of the thyroid left/right lobe, upper/lower part of the mediastinum, or elsewhere) at surgery on a per-lesion level, and if patients with one or several HPG(s) detected on imaging and confirmed by surgery findings on a per-patient level.

### 2.4. Statistical Analysis

The primary purpose of this analysis was to calculate summary sensitivity, specificity, positive likelihood ratio (PLR), negative likelihood ratio (NLR) and diagnostic odds ratios (DOR) with 95% confidence interval (CI) on a per-lesion level for the localization of HPG(s). A DOR can be calculated as the ratio of the odds of positivity in a disease state, relative to the odds of positivity in the non-disease state, with higher values indicating better discriminatory test performance [17]. We applied a bivariate random effects model to calculate the pooled sensitivity and specificity on a per-lesion level, then used the same model to plot summary receiver operating characteristic (SROC) curves and evaluate the areas under the curves (AUC). We only calculated pooled sensitivity and positive predictive value (PPV) on per-patient level, since some articles did not report FP and TN findings.

We assessed between-study heterogeneity of the data using the I-square index (I^2^) statistic and the Cochrane Q test, based on random-effects analysis [18]. To assess study-between heterogeneity, we performed the following subgroup analyses: (1) imaging procedure (4D-CT as first line vs. second line examination) on both analyses; and (2) parathyroid glands analysis (all glands vs. resected glands) on lesion-based level. Publication bias was evaluated by Deek’s test as previously described [19]. Data analyses were performed using the “Midas” and “Metaprop” modules in Stata software version 15.0 (StataCorp, College Station, TX, USA). Values were considered statistically significant if the two-sided *p* value was <0.05.

## 3. Results

### 3.1. Literature Search

Our search strategy resulted in a total of 238 studies, 76, 97, and 65 from PubMed, Embase, and Web of science, respectively. Among these, 100 duplicate articles as well 92 studies, including 33 not in the field of interest, 8 case reports, 31 abstracts, 15 reviews, one editorial, three congress and meetings, and one without full text were excluded. Among the remaining 46 articles, 16 [11,12,13,20,21,22,23,24,25,26,27,28,29,30,31,32] finally met the inclusion criteria and were subsequently included in our analysis. A flow diagram of eligible literature is shown in Figure 1.

### 3.2. Characteristics of the Included Studies

Basic characteristics of the 4D-CT studies are summarized in Table 1. Detailed technical aspects, including CT, as well as contrast and injection information, scan phase and imaging procedure are listed in Table 2. Among the 16 included studies, five described lesion-based analysis, seven reported patient-based analysis, whereas four were on both of the aforementioned analyses. Overall, these studies comprised a total of 1032 patients. Moreover, nine studies, including 710 patients and 2644 lesions on lesion-based analysis, and 11 studies comprising 503 patients on patient-based analysis allowed evaluation of 4D-CT’s role in localizing HPG(s), the diagnostic accuracy of 4D-CT are shown in Table 3. Of the 16 included studies, 13 and two had a retrospective and prospective design, respectively, whereas one study did not report the study design. Two studies focused on the group of patients operated on the second time or experiencing recurrence of the tumor, while the remaining 14 enrolled participants either without a history of prior neck surgery or with mixed type of the first and re-operation. Six and nine studies reported 4D-CT as a first- and second-line of examination, respectively, whereas one reported both examinations.

### 3.3. Quality Assessment

Results from methodological quality analysis, including patient selection, index test, reference standard, and flow and timing, are summarized in Figure 2. With regards to patient selection, nine studies showed an unclear risk of bias due to insufficient information of consecutive patient enrollment and time limitation, while one study revealed high risk of bias due to exclusion of multiple parathyroid glands. Based on index test, seven studies exhibited an unclear risk of bias owing to the fact that they did not mention whether or not the operators interpreted the images without reference standard, which might result in interpretation bias. We considered one study as high risk of bias, since not all interpretation of images were blinded. All studies revealed an unclear risk of bias in reference standard, owing to the fact that they did not mention whether or not the pathologist were blinded to prior clinical and imaging data, when they interpreted the histological results, which might result in verification bias. However, most of the studies were considered to have low applicability in the patient selection, index test and reference standard domains, therefore, we took all the studies into final analysis.

### 3.4. Summary of 4D-CT’s Diagnostic Performance

We regarded histological findings, as well as a combination of these and follow-up of biochemical resolution after surgery, as the reference standard. On a per-lesion analysis, sensitivity and specificity were used to assess the diagnostic performance. Conversely, on a per-patient analysis, the terms specificity and negative predictive value were not meaningful, since diagnosis was biologically confirmed for patients who were all assumed to have at least 1 HPG. Consequently, we only calculated pooled sensitivity and PPV as the metrics of diagnostic accuracy on a per-patient basis.

#### 3.4.1. Per-Lesion Level Analysis

The pooled sensitivity, specificity, PLR, NLR, and DOR of 4D-CT were 75% (95%CI: 66–82%), 85% (95%CI: 50–97%), 4.9 (95%CI: 1.1–21.3), 0.30 (95%CI: 0.19–0.45), and 17 (95%CI: 3–100), respectively. We summarized sensitivities and specificities of 4D-CT for localization of HPG(s) using forest plots as shown in Figure 3a. I^2^ values for sensitivity and specificity were 79% (Q = 38.13) and 94.7% (Q = 150.47), respectively, whereas AUC of 4D-CT was 81% (95%CI: 77–84%). The resulting SROC curve is shown in Figure 3b.

Subgroup analyses, performed by imaging procedure, revealed that four studies used 4D-CT as a second-line examination, and this resulted in a pooled sensitivity and specificity of 65% (95%CI: 54–75%) without significant heterogeneity (I^2^ = 22%, Q = 3.84) and 78% (95%CI: 3–100%) with significant heterogeneity (I^2^ = 94.9%, Q = 58.9). On the other hand, 5 studies reported use of 4D-CT as a first-line examination, and resulted in a higher pooled sensitivity (77%, 95%CI: 67–85%) and specificity (95%, 95%CI: 91–97%). Studies that reported 4D-CT as first-line examination revealed statistically significant heterogeneity in their sensitivity (I^2^ = 86%, Q = 28.54) and specificity (I^2^ = 93%, Q = 56.78) estimates. The AUC of the two subgroups were 66% (95%CI: 62–70%) and 94% (95%CI: 92–96%), respectively.

Subgroup analysis, in studies focusing on all parathyroid glands analysis for localization of culprit parathyroid gland(s), revealed pooled sensitivity, specificity and AUC values of 75% (95%CI: 66–83%), 94% (95%CI: 91–96%) and 94% (95%CI: 91–95%), respectively. Studies that used all parathyroid glands analysis for localization were statistically heterogeneous with regards to their sensitivity (I^2^ = 85.3%, Q = 33.92) and specificity (I^2^ = 87.8%, Q = 41.01) estimates.

Subgroup analysis, targeting study design, showed that eight retrospective studies had pooled sensitivity and specificity of 77% (95%CI: 70–83%) and 83% (95%CI: 38–98%), respectively, with an AUC of 81% (95%CI: 77–84%). Moreover, we observed heterogeneity in their sensitivity (I^2^ = 65.2%, Q = 20.12) and specificity (I^2^ = 95.1%, Q = 142.37). The diagnostic performance was similar to the overall results.

#### 3.4.2. Per-Patient Level Analysis

Patient-based analysis resulted in a pooled sensitivity of 81% (95%CI: 70–90%) and PPV of 91% (95%CI: 82–98%), with heterogeneity (I^2^ = 81.9% and I^2^ = 80.8%). A forest plot describing the sensitivity and PPV are shown in Figure 4. Subgroup analysis, performed by imaging procedure, showed that the pooled sensitivity were 77% (95%CI: 67–86%) and 86% (95%CI: 62–100%), and pooled PPV were 94% (95%CI: 86–100%) and 86% (95% CI: 68–98%) of 4D-CT as second- and first-line imaging modalities, respectively, with evidence of heterogeneity in their estimates of sensitivity (I^2^ = 59% and I^2^ = 91.2%) and PPV (I^2^ = 62.8% and I^2^ = 85.6%) in both subgroups. Furthermore, we did not perform subgroup analysis targeting study design (patient level), scan phase (three phases vs. four phases) as well as history of prior neck surgery (yes vs. no) and technical aspects, due to limited information from reported studies.

Further, Deek’s test showed no evidence of publication bias (*p* = 0.19 and *p* = 0.83) on a lesion- and patient-based level, the figures of funnel plot were supplemented in (Appendix A).

## 4. Discussion

The current meta-analysis revealed that 4D-CT has moderate diagnostic accuracy in patients with pHPT to a correct quadrant, based on both per-lesion and per-patient level, as evidenced by moderate pooled sensitivity, specificity and AUC. Moreover, subgroup analyses revealed that 4D-CT may improve diagnostic performance, as a first-line modality for in localizing patients with pHPT on a lesion-based level.

Previous meta-analyses have reported 76.1% (95%CI: 70.4–81.4%), 86% (95%CI: 80–90%), and 85% (95%CI: 69–94%) pooled sensitivity of US, 99mTc-sestamibi SPECT/CT and 4D-CT, respectively, during localization of HPG(s) in patients with HPT [14,15,33]. Parathyroid US and SPECT/CT are still references for preoperative localization in pHPT. Nevertheless, their sensitivities are particularly low in reoperative patients [34]. Additionally, the imaging modalities were reportedly negative or discordant in some clinical scenarios. However, 4D-CT has recently been shown to act as an alternative and emerging imaging modality. For example, Hamidi et al. [35] and Amadou et al. [20] found that 4D-CT exhibited 77.4 (lateralization) and 75% (quadrant or site) sensitivities during preoperative localization of HPGs in persistent pHPT (due to the unrecognized ectopic localizations during the first surgery, unknown multiglands and a negative preoperative imaging finding) [36,37,38]. Previous studies have also suggested that 4D-CT may play a role in preoperative localization in patients in whom conventional imaging scans failed to localize by conventional imaging scans, as evidenced by sensitivities ranging from 66 to 86% [20,23,29,30].

Wan et al. have reported the comparison of diagnostic value between 4D-CT and SPECT/CT head-to-head, their sensitivity and specificity of 4D-CT were slightly higher than the results of this study in a lesion-based analysis [15]. However, there are several methodological differences between the two meta-analyses. First, they only selected studies investigating 4D-CT with SPECT/CT head-to-head. Second, they performed meta-analysis on a lesion-based level without calculations based on a per-patient level. In an attempt to obtain more information evaluating the diagnostic value of 4D-CT, we included more studies. Moreover, we performed subgroup analysis to observe the effectiveness of 4D-CT for localization of HPG(s) in pHPT patients in different clinical settings.

With regards to lesion-based analysis, we found four studies that showed lower sensitivities (from 50 to 75%). Among them, one adopted a prospective and consecutive design, with a resulting sensitivity of 58%, which was in contrast to many other findings [20,32,39]. This might be, in part, due to the study design. The other three articles also demonstrated a relatively lower diagnostic accuracy, in which 4D-CT played a role as a second-line imaging modality for localizing HPG(s). Based on these findings, we performed subgroup analyses on study design and imaging procedure.

Subgroup analysis, by imaging procedure performed targeting lesions and patients, showed that second-line modality had lower diagnostic performance than first-line, with regards to both analyses for localization of HPG(s). This might be attributed to imaging results and the inexperience technicians operating inconvenient shooting techniques. Other factors may also change the effectiveness of 4D-CT, including positioning, large body habitus and position of HPG(s) [28,40,41]. Patients subjected to 4D-CT as a second-line examination always had negative or inconclusive first-line imaging results. Therefore, it was easy to increase the number of false-negatives. Of note, lesions in subgroup of 4DCT used as first-line imaging are significantly larger than those of second-line, the results of this group might be more convincing.

Since some of the studies included herein always assumed that each patient had 4 or more parathyroid glands [11,13,27,28,32], we performed subgroup analysis based on all parathyroid glands attributes and found high specificity (94%, 95CI: 91–96%) and AUC (94%, 95CI: 91–96%). However, other studies focusing on resected parathyroid lesions exhibited significantly lower specificities, possibly due to absence of analysis in negative findings on 4D-CT [30,31].

We did not perform subgroup analysis on prior neck surgery (yes vs. no) and scan phase (three phases vs. four phases), due to the limited number of history reports. In addition, among 16 included studies, there were only two studies reporting the diagnostic performance of 4D-CT for localization of pHPT patient with low/mild line hypercalcemia or parathyroid hormone (PTH), two studies focusing on persistent or recurrent pHPT, the published data was insufficient to evaluate the special group of patients. Notably, it is better to choose a protocol that allows minimal radiation dose without sacrificing diagnostic accuracy. For example, Raghavan et al. [42] demonstrated that arterial phase images (with a sensitivity of 91% and specificity of 82.9%) alone was adequate to obtain the diagnostic accuracy for parathyroid adenomas localization. In addition, Campbell et al. [43] suggested the two-phase computed tomography is as effective as 4D-CT for identifying enlarged parathyroid glands. Further research is needed to ascertain an ideal CT protocol.

Apart from conventional B-mode US, SPECT/CT, ultrasound elastography, a new tool for diagnosing parathyroid adenomas, has revealed a high sensitivity (93–100%) and specificity (90–95%) [44,45,46]. Additionally, 18F-choline and 11C-methionine positron emission tomography/computed tomography (PET/CT) are also useful diagnostic modalities for preoperative localization in pHPT patients [47,48]. Previous meta-analyses exhibited a pooled sensitivity 92% (95CI: 88–96%) of 18F-choline PET and 81% (95CI: 74–86%) of 11C-Methionine PET/CT on a per-patient level [49,50]. Furthermore, 3T magnetic resonance imaging (MRI) showed an excellent sensitivity 97.8% (95CI: 92.3–99%) and specificity 97.5% (95CI: 97–100%) for preoperative localization of parathyroid adenomas [51]. However, up to now as no studies to compare the diagnostic accuracy of 11C-Methionine, 18F-choline PET, MRI with 4D-CT, further investigations are needed.

This meta-analysis had some limitations. Firstly, as shown by the QUADAS 2 tool, an unclear and a high risk of bias, arising from patient selection, index text and reference standard, should be took into consideration. It was difficult to avoid patient selection bias owing to the fact that only two studies had a prospective design, while a majority of the rest were retrospective. Furthermore, whether reviewers/pathologists were blinded to histological/imaging results or not can change the diagnostic accuracy to some extent. The results of the current meta-analysis should therefore be interpreted with caution. Secondly, some studies only exhibited diagnostic accuracy based on a single lesion, and excluded patients with multiple parathyroid glands. This could have lowered sensitivity. Thirdly, not all reference standards combined histological findings with follow up. Fourthly, a previous study reported that radio-labelled choline PET showed excellent diagnostic performance in the detection of HPG(s) in HPT patients. However, in our case, only two studies reported diagnostic accuracy of 4D-CT and F18 choline-PET/CT on lesion-based analysis, which was not enough to evaluate the diagnostic performance of the two imaging modalities.

## 5. Conclusions

Overall, 4D-CT exhibited moderate sensitivity and specificity in both patient- and lesion-based analysis of preoperative localization of HPG(s). Its diagnostic performance may improve when it was used as a first-line modality, nevertheless, necessitating further investigation to compare the diagnostic accuracy of 4D-CT with other imaging modalities to confirm our observation.

## Figures and Tables

**Figure 1 diagnostics-11-00664-f001:**
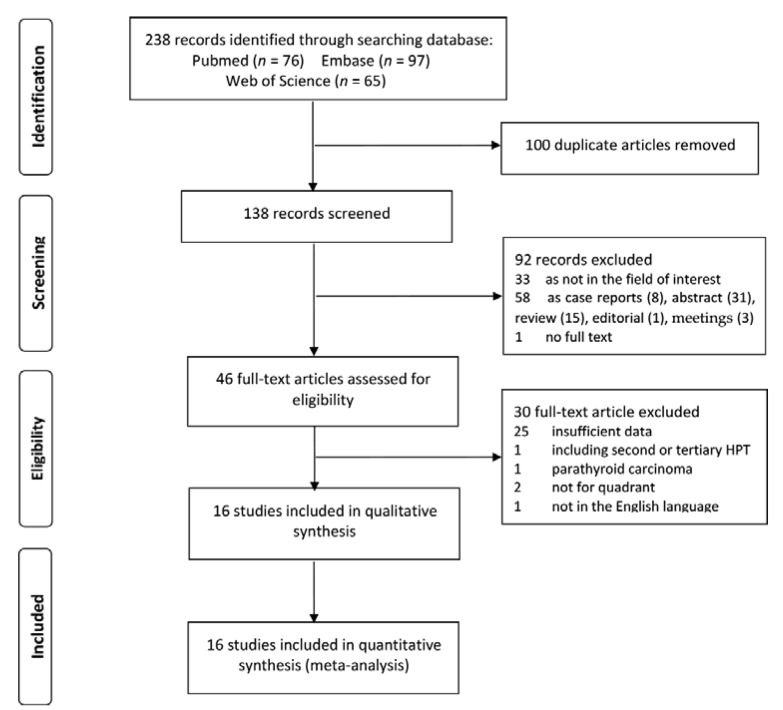
Flow algorithm of selection for eligible studies.

**Figure 2 diagnostics-11-00664-f002:**
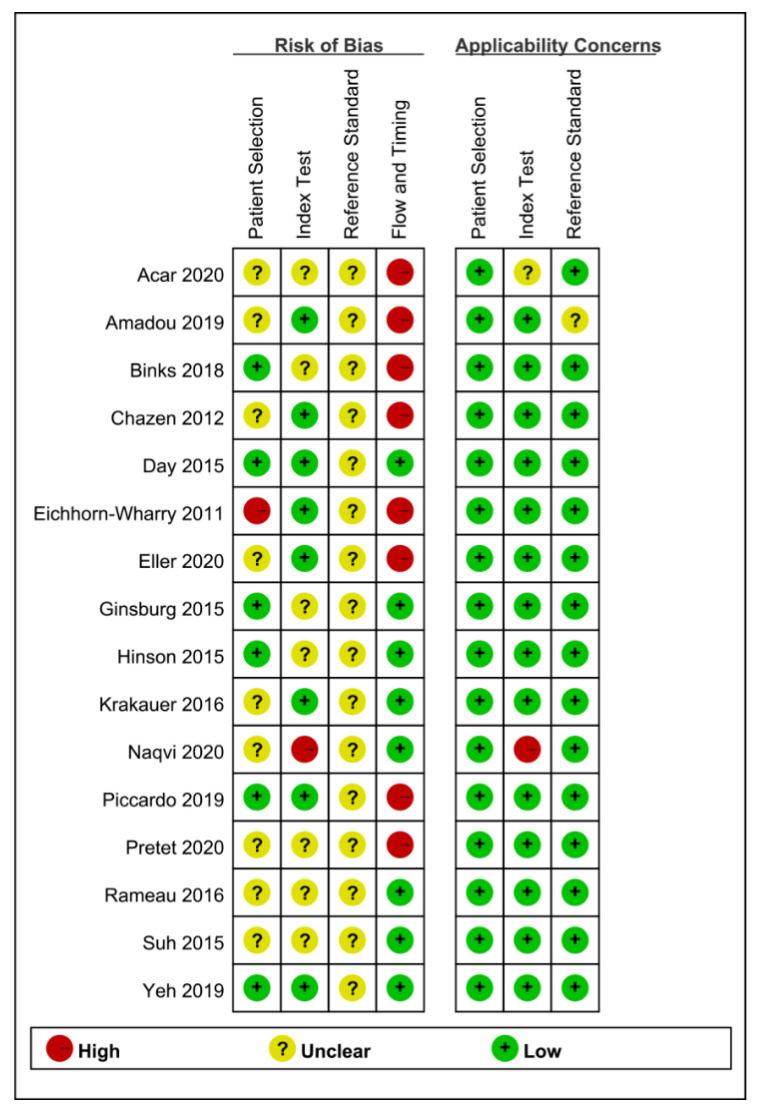
Methodology of included studies by Quality Assessment of Diagnostic Accuracy Studies 2 tool (QUADAS-2).

**Figure 3 diagnostics-11-00664-f003:**
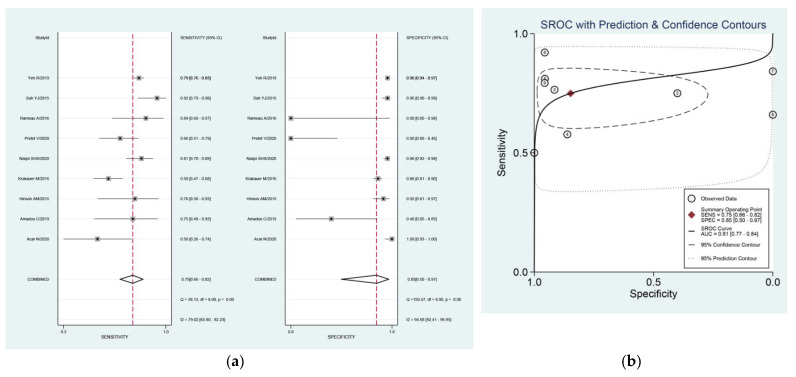
(**a**) Forest plot of pooled sensitivity and specificity and (**b**) summary receiver-operating characteristic (ROC) curves of four-dimensional computed tomography (4D-CT) for assessing localization of hyperfunctioning parathyroid glands (HPGs) in primary hyperparathyroidism (pHPT) patients on a per-lesion level. The area under the ROC curves of 4D-CT was 0.81.

**Figure 4 diagnostics-11-00664-f004:**
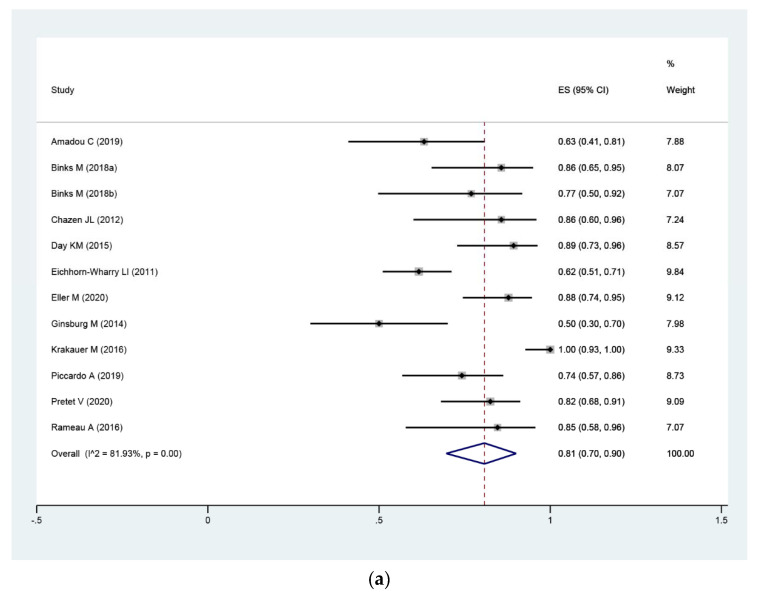
Forest plot of pooled sensitivity (**a**) and positive predictive value (PPV) (**b**) of 4D-CT for localization of HPGs in pHPT patients on a per-patient level, including 0.95 confidence interval (CI).

**Table 1 diagnostics-11-00664-t001:** Basic information of included study characteristics.

Source	Country	Study Design	Patient No.	Age	F/M	Prior Neck Surgery	Reference Standard
Acar N (2020)	Turkey	R	17 (33)	59.5	16/1	No	histology + follow up
Amadou C (2019)	France	R	19 (29)	NA	NA	Yes	histology, US-guided FNA
Binks M (2018)	Australia	R	13, 21 (165) *	NA	NA	NA	histology + follow up
Chazen JL (2012)	USA	R	25 (35)	NA	NA	Yes	histology
Day KM (2015)	USA	R	37 (37)	63	NA	Yes	histology + IOPTH
Eichhorn-Wharry LI (2011)	USA	NA	135 (135)	59.2 ± 13	109/26	Yes	histology
Eller M (2020)	USA	R	51 (100)	NA	NA	No	operation + histology
Ginsburg M (2015)	USA	R	22 (28)	NA	NA	Yes	histology
Hinson AM (2015)	USA	R	19 (19)	66	16/3	No	histology + follow up
Krakauer M (2016)	Denmark	P	91 (91)	66	67/24	No	surgical findings + histology
Naqvi SHS (2020)	USA	R	68 (68)	65.5	NA	NA	surgical findings + histology
Piccardo A (2019)	Italy	P	31 (44)	NA	NA	NA	histology + follow up
Pretet V (2020)	France	R	44 (50)	NA	NA	Yes	histology + IOPTH + follow up
Rameau A (2016)	USA	R	14 (14)	57.6	12/2	No	surgical reports + histology
Suh YJ (2015)	Korea	R	38 (38)	55.8 ±13.2	27/11	No	surgical reports + histology
Yeh R (2019)	USA	R	400 (400)	61 ± 14	319/81	No	surgical reports + histology

NOTE: NA—not available, No.—Number R—Retrospective P—Prospective, Patient No.—included patients (total patients), IOPTH—Intraoperative parathyroid hormone. *—This study reported 4D-CT was used as both first and second line examination for two different group of patients.

**Table 2 diagnostics-11-00664-t002:** Technical aspects.

Source	Machine Model	Detector	Product	Injection Rate (mL/s)	Dose	Slice Thickness	Scan Phase	Imaging Procedure
Acar N (2020)	Siemens Somatom	128	Iodine	NA	1 mL/kg	NA	4	2
Amadou C (2019)	NA	NA	NA	NA	NA	NA	3	2
Binks M (2018)	NA	NA	NA	NA	NA	NA	3	both
Chazen JL (2012)	GE LightSpeed	16, 64	Omnipaque	4	2 mL/kg 120 mL max	1.25 mm	3	1
Day KM (2015)	GE LightSpeed	64	Omnipaque	3	100 mL	0.625 mm	both	2
Eichhorn-Wharry LI (2011)	GE 16, VCT Lightspeed	16	Optiray	NA	100 mL	1.25 mm	3	1
Eller M (2020)	NA	NA	NA	NA	NA	NA	NA	2
Ginsburg M (2015)	Philips Brilliance	64, 256	Omnipaque	4	120 mL	0.9 mm	4	2
Hinson AM (2015)	Philips	64	nonionic contrast	3	75 mL	NA	4	2
Krakauer M (2016)	Philips Skylight	16	Omnipaque	3.5	100 mL	2 mm	3	1
Naqvi SHS (2020)	NA	NA	NA	NA	NA	NA	NA	1
Piccardo A (2019)	NA	16	Iohexol, Iodine	3.3–4	80–100 mL 350 mg	1.25 mm	3	2
Pretet V (2020)	Philips Biograph	128	Iomeron	2.5–3	75 mL	1 mm	4	2
Rameau A (2016)	GE LightSpeed, VCT	16, 64	Omnipaque	4	100–120 mL	1.25 mm	3	1
Suh YJ (2015)	Philips Brilliance	64	Xenetics	NA	90 mL	2 mm	4	1
Yeh R (2019)	Siemens Symbia T	16	Omnipaque	4	75 mL	1 mm	3	1

NOTE: Imaging procedure: 1—first line, 2—s line, Scan phase: 3—three phases, 4—four phases.

**Table 3 diagnostics-11-00664-t003:** Diagnostic accuracy of four-dimensional computed tomography (4D-CT) for localization of hyperfunctioning parathyroid glands (HPGs).

Source	Patient No.	True Positives	False Positives	False Negatives	True Negatives	LB	PB
Acar N (2020)	17 (33)	9	0	9	51	√	
Amadou C (2019)	19 (29)	12	3	4	2	√	
		12	0	7	0		√
Binks M (2018a)	21 (165)	18	1	3	0		√
Binks M (2018b)	13 (165)	10	0	3	0		√
Chazen JL (2012)	25 (35)	12	0	2	11		√
Day KM (2015)	37 (37)	25	9	3	0		√
Eichhorn-Wharry LI (2011)	135 (135)	53	11	33	NA		√
Eller M (2020)	51 (100)	36	5	5	5		√
Ginsburg M (2015)	22 (28)	10	0	10	2		√
Hinson AM (2015)	19 (19)	13	5	4	54	√	
Krakauer M (2016)	91 (91)	56	37	41	230	√	
		49	36	0	0		√
Naqvi SHS (2020)	68 (68)	60	9	14	194	√	
Piccardo A (2019)	31 (44)	23	0	8	0		√
Pretet V (2020)	44 (50)	33	6	17	0	√	
		33	4	7	0		√
Rameau A (2016)	14 (14)	16	1	3	0	√	
		11	1	2	0		√
Suh YJ (2015)	38 (38)	35	5	3	109	√	
Yeh R (2019)	400 (400)	414	47	108	1031	√	

NOTE: LB—lesion-based, PB—patient-based.

## Data Availability

All analyses were based on previously published studies. Data sharing is not applicable to this article.

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
