# Peer review of "Diagnostic Role of Four-Dimensional Computed Tomography for Preoperative Parathyroid Localization in Patients with Primary Hyperparathyroidism: A Systematic Review and Meta-Analysis"

_diagnostics, 2021, doi:10.3390/diagnostics11040664_

Round 1

Reviewer 1 Report

The paper ”Diagnostic role of four-dimensional CT…” by Sun et al. presents a comprehensive systematic review in three major databases and respective meta-analyses thereon. The statistical analysis is pretty much state of the art, the application of QUADAS-2 is highly appreciated and informative. Please find detailed comments below.

Major

1) l.16, l.71, l.118-122, l.151-153: The authors argue that only sensitivity on a per-patient level could be shown. First of all, corresponding analyses for the positive and negative predictive values can and should be performed on the per-lesion level as well. Secondly, on the per-patient level, the positive predictive value can be added to the analysis of sensitivity. Thirdly, though it is evident that data on per-patient level was slightly more scarce than data on per-lesion level, it is the per-patient level which at the end of the day counts. Usually, the per-patient level analysis is primary, the per-lesion level analysis supplementary and secondary.

2) l. 37-38: The authors state that neck ultrasound and 99mTc-sestamibi SPECT/CT are the most commonly used. How about Methionin PET (see, for instance, doi: 10.1097/MNM.0000000000000216; PMID: 25244350 and doi: 10.1177/0284185116661878; PMID: 27589877).

3) l.55-56: As the present study aimed at evaluating diagnostic performance of 4D-CT for the localization of HPG(s), definitions for true positives, false negatives, true negatives and false positives should be given in the Methods section for both the per-lesion and per-patient level. This is not trivial.

4) l.61-65: The search terms are only given broadly. Please add specific search strings / MeSH terms and their combinations as Supplemental Material to transparently report your conducted search and to make it reproducible by and for the reader, thanks. In l.64-65, logical AND and OR should be emphasized by appropriate parentheses, for instance, “(diagnostic accuracy) OR sensitivity OR specificity” – the last term “sensitivity AND specificity” is redundant as already covered by the respective OR connection before.

5) l.88-90: I guess the sentence should end “on a per-lesion level for the localization of HPG(s).” in order to highlight that the study is not about identification yes/no, but localization of (somewhere yet to be identified and localized) lesions.

6) l.93: How was the intra-patient cluster structure of lesions acknowledged in the bivariate random effects model? There have been probably by far more lesions than patients in the included studies.

7) l.95-97: See comment above regarding positive predictive value on a per-patient basis. Moreover, please add a table summarizing true positives, false negatives, true negatives and false positives on a both per-lesion and per-patient level to the extent possible. Though some studies did not report true negatives and false positives, you may still consider and be able to produce a forest plot for specificity and negative predictive value on a smaller number of studies.

8) l.98: Apart from the I-squared statistic regarding heterogeneity between studies, funnel plots are common to illuminate potential publication bias. These can be added as Supplemental Material as well, but they should definitely be part of any thorough meta-analysis like yours.  

9) Figure 2: It is striking to see the consistency of “unclear risk of bias” with respect to the reference standard and the prominent risk of bias regarding flow and timing.

Minor

10) l.23: “P=0.00” should read “P<0.001”. A peculiarity of respective STATA outputs is that “P=0.00” reads in the plots (see, for instance, Figure 4, last line) when actually “P<0.001” is meant.

11) l.32: “have been shown to result”

12) l.33: “short operative time and hospital stay” – no comma

13) l.46 “over time”

14) l.52: please add the range for the number of patients contributing to these three studies as these substantiate the level of evidence behind the wide ranges for sensitivity and specificity.

15) l.53: please add a 95% CI for the pooled sensitivity given as 89% and/or give number of studies and respective patients behind this number.

16) l.85-86: if the very same two authors performed the screening and these are co-authors, please add their initials for identification.

17) l.99: between-study heterogeneity

18) Figure 1: The top box should contain the total number of hits of N=238, e.g. “238 records identified through…”; second box: “100 duplicate articles removed”

19) l.122-123: 13 and 2 had a retrospective and prospective design

20) l.124-125: I guess you do not mean “group of reoperative and recurrent people” but “group of patients operated on the second time or experiencing recurrence of the tumour”

21) Table 1: please explain in the table legend what the entries in the column “Patient No.” mean, e.g. “17(33)”; why are three numbers given in the third line here (“13,21(165)”); table legend: R-Retrospective

22) l.135: flow and timing (not: flow timing)

23) l.138-140: you can specify the described biases by commonly used terms, see Hall, Kea and Wang in their 2-paper series from 2019: doi: 10.1136/emermed-2019-208446, PMID: 31302605 and doi: 10.1136/emermed-2019-208447, PMID: 31221671.

24) l.141: were blinded

25) l.141-142: It is striking that all studies revealed an unclear risk of bias in the reference standard. Here, it becomes eminent to which extent the application of QUADAS-2 contributes! By the way, please add ‘of bias’ to risk, i.e. risk of bias.

26) l.142-143: “owing to lack of information with regards to histological results without prior clinical and imaging data” is unclear to me – what do you mean by histology without knowledge of prior clinical and imaging data?

27) l.168-169: with / without significant heterogeneity – I doubt that heterogeneity actually can be totally absent.

28) l.184: replace “While Deek’s test further showed” by “Further, Deek’s test showed”

29) l. 215: delete “be”

30). l.215-216: it is not the patients who fail to localize; rephrase, for instance, as “in patients in whom conventional imaging scans failed to localize”

31) l.245: Further research is needed (or: warranted)

32) l.260: delete “was” – its diagnostic performance improved

Reviewer 2 Report

Despite a metanalysis on the diagnostic performance of 4D CT could be useful for clinical practice, the main limitation of this study is that Authors did not compare this technique with other imaging modalities, as 99mTc-MIBI SPECT/CT; 18F-Fluorocholine PET/CT, ultrasound, MRI. Authors should describe also this diagnostic tecniques (see for example, Argirò et al. PMID:29736849; Isidori et al. PMID 27709473; Cuderman et al, 31562221).

Moreover, Authors should discuss their finding according to the results by a recent metanalysis (Wan QC, Li JF, Tang LL, Lv J, Xie LJ, Li JP, Qin LP, Cheng MH. Comparing the diagnostic accuracy of 4D CT and 99mTc-MIBI SPECT/CT for localizing hyperfunctioning parathyroid glands: a systematic review and meta-analysis. Nucl Med Commun. 2021 Mar 1;42(3):225-233. doi: 10.1097/MNM.0000000000001331. PMID: 33306636)

Introduction:

- Line 30: Authors stated that “Parathyroidectomy represents the best line of action for management of parathyroid adenoma(s)”. Please describe that surgical intervention is indicated only in a proportion of patients according to specific criteria (Bilezikian, 2014, PMID: 25162665).

- Line 52-54: Please cite also a recent systematic review and metanalysis (Wan QC, Li JF, Tang LL, Lv J, Xie LJ, Li JP, Qin LP, Cheng MH. Comparing the diagnostic accuracy of 4D CT and 99mTc-MIBI SPECT/CT for localizing hyperfunctioning parathyroid glands: a systematic review and meta-analysis. Nucl Med Commun. 2021 Mar 1;42(3):225-233. doi: 10.1097/MNM.0000000000001331. PMID: 33306636). Please also discuss the findings of this article in the discussion section.

Materials and Methods:

- Please specify if the systematic review and metanalysis has been performed according to PRISMA guidelines and if it has been registered in PROSPERO (International prospective register of systematic reviews)

Results:

- Table 1: please specify if the articles have been used for lesion-based analysis or patient-based analysis or both.

- Legend to Table 1: NA is missing from the legend; probably “respective” should be “retrospective”.

Discussion:

- Authors should discuss more thoroughly what is missing in the published articles in order to provide information on future research topics needed.

- Authors should compare more carefully articles using 4D CT as first-line modality or not in order to suggest a possible explanation of the increased performance accuracy as first line examination, which seems difficult to explain.

- In the limitation sections, authors should criticize if the quality of the studies (reported at point 3.3. Quality assessment) could affect results of the metanalysis.  

Conclusions:

- Conclusions are probably too small. Authors should be careful in assessing that “diagnostic performance was improved when it was used as a first-line modality” since the use as first line imaging modality could be considered a recommendation but the article did not compare diagnostic accuracy of 4D CT with other imaging modalities

 - Line 263: please change “its” with “it”.

Reviewer 3 Report

In the manuscript "Diagnostic role of four-dimensional computed tomography for preoperative parathyroid localization in patients with primary hyperparathyroidism: a systematic review and meta-analysis", the authors performed a systematic review and a meta-analysis in order to establish the diagnostic performance of four-dimensional computer tomography in localization of preoperative parathyroid glands in patients with primary hyperparathyroidism. However, major criticisms are present, as follows: - In the introduction part the clinical setting of primary hyperparathyroidism should be briefly discussed, like incidence and prevalence in hyperparathyroidism; - In the material and methods section line 66-67 the authors state that “Neither start date nor language restrictions were included in our search” and in line 75-77 they state that “Articles were excluded if they were duplicate publications, case reports, abstracts, review articles, conference/meeting papers, lacked full text or were not written in English. Moreover, articles that reported insufficient data as well as those that only described diagnosis of HPG(s) to a correct side were also excluded.” Please make this clear, as it is very confusing on how did you selected the papers. - In line 85-86 – the authors discuss about discrepancies resolved by discussion. Could you please state what discrepancies you encountered? - In the results section, I find that figure 1 is not necessary, as the algorithm is already discussed and please do a better description and classification of the studies, as the characteristics of the included studies are very hard to follow. - As the title states, the evaluation of four-dimensional computed tomography should evaluate only primary hyperparathyroidism, but in the result section, point 3.2, the authors discuss about papers on reoperative and recurrent hyperparathyroidism, please explain on what basis you included patients with recurrent disease. Is it parathyroid hyperplasia disease or is it secondary hyperparathyroidism? - On table 2, please comment if you found any differences between patients who had prior neck surgery, what are the causes of their recurrence and if you found any significant differences in the four-dimensional computed tomography evaluation between the group who was initially evaluated on 4DCT before surgery versus the group who had an reintervention, does the specificity, sensitivity and accuracy modify between the two groups? - In the results section, the authors mention that they had studies with unclear risk,but did not mention if they took the papers into consideration on the final analysis. - In the discussion part, you did not compare the discriminative power of 4D CT with ultrasound and especially with elastography. Important papers on elastography have been published and should be taken into consideration in the discussion part, as it can have an important role in the localization of parathyroid adenomas, as stated in an important paper published in 2020 - "Shear Wave Elastography versus Strain Elastography in Diagnosing Parathyroid Adenomas", International Journal of Endocrinology, vol. 2020, Article ID 3801902, 11 pages, 2020. -Conclusions should be better stated and I consider that this paper should be considered a review rather than a meta-analysis.

Reviewer 4 Report

Authors intensively reviewed the clinical effectiveness of parathyroid 4D CT. I think that they performed meaningful study and it is worthy to publication in Diagnostics journal. 

Author Response

Authors intensively reviewed the clinical effectiveness of parathyroid 4D CT. I think that they performed meaningful study and it is worthy to publication in Diagnostics journal. 

Response: Thanks for your comments.

Round 2

Reviewer 2 Report

Please note that line numbers reported in the response to reviewers file do not match with the revised manuscript and this has complicated the second revision of the manuscript.

Minor comments:

- Line 42: please change “as well as medical cost” with “as well as reduced medical cost”

- Line 153: please check “NA-missing from the legend”, since probably is NA-not available

Author Response

Response to Reviewer 2 Comments

Point 1

Line 42: please change “as well as medical cost” with “as well as reduced medical cost”

Response1: Firstly, thanks a lot for your review once a time. We have changed “as well as medical cost” with “as well as reduced medical cost” in line 44.

Point 2

Line 153: please check “NA-missing from the legend”, since probably is NA-not available.

Response 2: We have changed “NA-missing from the legend”with “NA-not available”, since several studies explained NA of “not available” in line 158.

Reviewer 3 Report

The paper has a high quality and the paper is reproducible. Accept in current form.
